# Baseline High-Resolution CT Findings Predict Acute Exacerbation of Idiopathic Pulmonary Fibrosis: German and Japanese Cohort Study

**DOI:** 10.3390/jcm8122069

**Published:** 2019-11-24

**Authors:** Chihiro Hirano, Shinichiro Ohshimo, Yasushi Horimasu, Hiroshi Iwamoto, Kazunori Fujitaka, Hironobu Hamada, Nobuoki Kohno, Daisuke Komoto, Kazuo Awai, Nobuaki Shime, Francesco Bonella, Josune Guzman, Hilmar Kühl, Ulrich Costabel, Noboru Hattori

**Affiliations:** 1Department of Molecular and Internal Medicine, Graduate School of Biomedical and Health Sciences, Hiroshima University, 1-2-3 Kasumi, Minami-ku, Hiroshima 734-8551, Japan; 2Department of Emergency and Critical Care Medicine, Graduate School of Biomedical and Health Sciences, Hiroshima University, Hiroshima 734-7551, Japan; 3Hiroshima Cosmopolitan University, Hiroshima 734-0014, Japan; 4Department of Diagnostic Radiology, Graduate School of Biomedical and Health Sciences, Hiroshima University, Hiroshima 734-8551, Japan; 5Department of Diagnostic Radiology, National Hospital Organization Kure Medical Center and Chugoku Cancer Center, Hiroshima 737-0023, Japan; 6Interstitial and Rare Lung Disease Unit, Department of Pneumology, Ruhrlandklinik, University Hospital Essen, 45239 Essen, Germany; 7General and Experimental Pathology, Ruhr University Bochum, 44801 Bochum, Germany; 8Department of Diagnostic and Interventional Radiology and Neuroradiology University Hospital Essen, 45147 Essen, Germany

**Keywords:** idiopathic pulmonary fibrosis (IPF), acute exacerbation (AE), high-resolution computed tomography (HRCT), ethnicity, ground glass opacity (GGO)

## Abstract

Acute exacerbation of idiopathic pulmonary fibrosis (AE-IPF) is a major cause of morbidity and death in IPF. However, sensitive predictive factors of AE-IPF have not been well-investigated. To investigate whether high-resolution computed tomographic (HRCT) abnormalities predict AE-IPF in independent ethnic cohorts, this study included 121 patients with IPF (54 German and 67 Japanese; mean age, 68.5 ± 7.6 years). Two radiologists independently visually assessed the presence and extent of lung abnormalities in each patient. Twenty-two (18.2%) patients experienced AE-IPF during the follow-up. The incidence of AE-IPF was significantly higher in the Japanese patients (*n* = 18, 26.9%) than in the German patients (*n* = 4, 7.3%, *p* < 0.01). In the Kaplan–Meier analysis, patients with a larger extent of ground glass opacity (GGO), fibrosis, and traction bronchiectasis experienced an earlier onset of AE-IPF (*p* = 0.0033, 0.0088, and 0.049, respectively). In the multivariate analysis, a larger extent of GGO and fibrosis on HRCT were independent predictors of AE-IPF (*p* = 0.026 and 0.037, respectively). Additionally, Japanese ethnicity was independently associated with the incidence of AE-IPF after adjustment for HRCT findings (*p* = 0.0074). In conclusion, a larger extent of GGO and fibrosis on HRCT and Japanese ethnicity appear to be risk factors for AE-IPF.

## 1. Introduction

Idiopathic pulmonary fibrosis (IPF) is characterized by progressive fibrosis of the lung with unknown etiology resulting in a continuous deterioration of pulmonary function and poor outcomes. The clinical course of IPF is usually progressive, and disease progression can occur at different rates. There are even cases that seem to stabilize over many years. Some patients experience abrupt deterioration in their respiratory status with no identifiable cause, which is termed acute exacerbation of IPF (AE-IPF). AE-IPF is one of the most important causes of death in patients with IPF [1,2,3].

There have been several studies that identified clinical risk factors for AE-IPF, including poor pulmonary function [3,4,5], rapid decline in vital capacity (VC) [6,7,8], and pulmonary hypertension [9]. However, pulmonary function tests require the maximum effort of patients, which may occasionally be difficult in patients with severe respiratory failure. Right-heart catheterization for confirming pulmonary hypertension is invasive. Several biomarkers, such as KL-6, Surfactant protein-D, matrix metalloproteinases, and CC Chemokine Ligand-18 have also been investigated as predictors of AE-IPF, but they have not yet been validated [10]. High-resolution computed tomography (HRCT) has great potential as a noninvasive tool, not only for diagnostic purposes but also for predicting prognosis [11,12,13]. However, the association between HRCT characteristics and the risk of AE-IPF has not been extensively explored. 

It has been suggested that Asian patients are more likely to experience AE-IPF than other ethnic patients [14]. Recently, Reichmann et al. demonstrated that Caucasian patients with IPF experienced less AE-IPF than non-Caucasian [8], while there was no difference in the risk of AE-IPF between Asian and White patients in the trial [15]. It is, therefore, still controversial whether ethnicity affects the incidence of AE-IPF. 

The aim of our study was to investigate the association between HRCT findings and the risk of AE-IPF in a well-defined international cohort.

## 2. Methods

This study was approved by the Institutional Review Board in Ruhrlandklinik (IRB 06-3170) and Hiroshima University Hospital (IRB 1964). Written informed consent was not obtained from each participant, because this was a retrospective study using only clinical and HRCT data from daily practice, which is part of the standard of care for examinations.

### 2.1. Subjects

We reviewed the medical records of all patients who were diagnosed as having IPF in the Hiroshima University Hospital (Hiroshima, Japan) from 2003 to 2012, and the Ruhrlandklinik (Essen, Germany) from 2006 to 2012. All patients underwent HRCT scanning. The diagnosis of IPF was made according to the American Thoracic Society/European Respiratory Society consensus classification 2002 [16], and those with chronic hypersensitivity pneumonia, connective tissue disease-related interstitial lung disease, or other known causative interstitial lung diseases were excluded. We excluded patients who presented AE-IPF at the time of diagnosis of IPF, or developed AE-IPF within one month from the first diagnosis (*n* = 9). The remaining 121 patients were included in our study (67 Japanese patients and 54 German patients). The median durations of follow-up for the Japanese and the German IPF patients were 36 months and 31 months, respectively.

### 2.2. Definition of AE-IPF

Diagnosis of AE-IPF was made according to a modification of the criteria defined by Collard et al. [17]: (1) Unexplained worsening or development of dyspnea within one month; (2) HRCT with new bilateral ground-glass abnormality and/or consolidation superimposed on a background reticular or honeycomb pattern; and (3) exclusion of alternative causes, including infection, left heart failure, and an identifiable cause of acute lung injury. 

### 2.3. Pulmonary Function Tests

Physiological assessments included measurements of thoracic gas volume, vital capacity (VC), forced expiratory volume in 1 s, and single-breath diffusing capacity of the lung for carbon monoxide (DLco) according to the American Thoracic Society/European Respiratory Society guidelines [18]. 

### 2.4. Radiographic Assessments

HRCT scans were performed at the time of stable disease and obtained at the end of inspiration using a variety of scanners. The protocols required 1–2 mm collimation sections reconstructed at 1 or 2 cm intervals. Due to the generation of CT scanners used, the examinations were not performed as low-dose CT, and image reconstruction was done without iterative reconstruction. All images were anonymized using Digital Imaging Communications in Medicine at standard window settings for visualization of the lung parenchyma (window level −600 HU, window width 1500 HU). The images were visually reviewed by two radiologists (one Japanese radiologist, DK, with 10 years of experience, and one German radiologist, HK, with 20 years) who were blinded to all clinical and functional data. The lung was divided into six zones (upper zone: above the tracheal carina, middle zone: between upper and lower zone, lower zone: below the inferior pulmonary vein). The reviewers evaluated the extent of the emphysema, ground glass opacity (GGO), and fibrosis area, which consisted of reticulation and honeycombing. They also evaluated the presence or absence of traction bronchiectasis in each zone. The extent of emphysema and lung parenchymal abnormalities was estimated to the nearest 5% for each of the six zones. The score of each finding was calculated by averaging the scores of the six zones. We defined the extent of traction bronchiectasis as the number of zones with the presence of traction bronchiectasis. The average of the two radiologists’ evaluations was used for the analysis. The findings were defined according to the Fleischner Society’s glossary [19].

### 2.5. Statistical Analysis

Individual variables for two groups were analyzed by Mann–Whitney’s *U*-test or chi-square test, as appropriate. The inter-observer agreement was assessed using the intraclass correlation coefficient. Intraclass correlation coefficients of 0.61–0.80 were interpreted as substantial and 0.81–1.00 as excellent agreement, as previously described [20]. Correlations between variables were determined by using Spearman correlation coefficients. The cut-off values were defined as the values with the highest Youden index (i.e., sensitivity+specificity-1) on a receiver-operating characteristic (ROC) curve for distinguishing patients who developed AE-IPF from patients without developing AE-IPF during follow-up. The probability of developing AE-IPF was estimated with the Kaplan–Meier method, and the differences in AE-free rates were evaluated by the log-rank test. The multivariate analysis of predictive factors for AE-IPF was done using the Cox regression hazard model. Patients with no AE-IPF events were censored at the date of death, loss of follow-up, or completing 3 years follow-up. The available-case analysis was applied for missing data. In all statistical analyses, *p* values less than 0.05 were regarded as significant. All statistical analyses were performed with EZR (Saitama Medical Center, Jichi Medical University, Saitama, Japan) [21], which is a graphical user interface for R (version 3.1.1, The R Foundation for Statistical Computing, Vienna, Austria). 

## 3. Results

### 3.1. Patients’ Characteristics 

Table 1 demonstrates the patients’ demographics and clinical characteristics. There were no significant differences in age, gender, smoking status, and pulmonary functions at baseline between German and Japanese patients with IPF. Twenty-two (18.2%) patients experienced AE-IPF during the three years of follow-up (median, 33 months), of which nine patients (40.9%) died within three months of AE-IPF. The incidence of AE-IPF was significantly higher in the Japanese patients than in the German patients (n = 18, 26.9% vs. n = 4, 7.3%, *p* < 0.01). In this study, no AE-IPF was induced by surgery, lung biopsy, bronchoalveolar lavage, or chemotherapy.

### 3.2. Inter-Observer Agreement

The inter-observer agreement was substantial for the extent of emphysema, GGO, fibrosis area (Intraclass correlation coefficient, ICC = 0.70 *(*0.60–0.78*)*, ICC = 0.75 *(*0.66–0.82*)*, ICC = 0.73 *(*0.63–0.80*)*), and also for the extent of traction bronchiectasis (ICC = 0.68 *(*0.58–0.77*)*) and for the two components of the fibrosis area: reticulation (ICC = 0.74 *(*0.65–0.81*)*) and honeycombing (ICC = 0.62 *(*0.50–0.72*)*). 

### 3.3. Correlation between HRCT Findings 

Firstly, we analyzed the correlation between the various HRCT characteristics. The extent of GGO was weakly correlated with the extent of fibrosis area (*r* = 0.22; *p* = 0.017). The extent of fibrosis area was moderately correlated with the extent of traction bronchiectasis (*r* = 0.58; *p* < 0.01), There was no significant correlation between the other HRCT findings.

### 3.4. HRCT Findings and Pulmonary Function

Secondly, we analyzed the correlation between the various HRCT findings and pulmonary function. The extent of GGO, fibrosis area, and traction bronchiectasis were inversely correlated with %VC (*r* = −0.23; *p* < 0.01, *r* = −0.37; *p* < 0.01, and *r* = −0.29; *p* < 0.01, respectively), and also inversely correlated with %DLco (*r* = −0.29; *p* < 0.01, *r* = −0.57; *p* < 0.01, and *r* = −0.30; *p* < 0.01, respectively). The extent of emphysema was inversely correlated with %DLco (*r* = −0.26; *p* < 0.01). 

### 3.5. HRCT According to Ethnicity

HRCT findings according to ethnicity are shown in Table 2. The extent of GGO and emphysema were significantly larger in the Japanese patients than in the German patients, while the extent of fibrosis and the zones of traction bronchiectasis were significantly larger in the German patients than in the Japanese patients.

### 3.6. Association between HRCT Findings and Risk of AE-IPF

ROC analyses showed that the cut-off levels for predicting development of AE-IPF were 4.5% for the extent of GGO, 18.7% for the extent of fibrosis area, 8.3% for the extent of emphysema, and more than five zones for traction bronchiectasis. In the Kaplan–Meier analysis, patients with a larger extent of GGO, fibrosis area, and traction bronchiectasis experienced an earlier onset of AE-IPF than those with a lower extent of abnormalities (*p* = 0.0033, 0.0088, and 0.049, respectively) (Figure 1). In the univariate Cox-proportional hazards analysis, Japanese ethnicity, low %VC, and a larger extent of GGO and fibrosis area on HRCT were significantly associated with the risk of AE-IPF (Table 3). Regarding the treatment regimen, the use of corticosteroids, immunosuppressive agents, as well as pirfenidone had no impact on the occurrence of AE-IPF. In the multivariate analysis, Japanese ethnicity and a larger extent of GGO and fibrosis area on HRCT were independently associated with the risk of AE-IPF after adjustment for age, sex, and %VC (Table 4).

## 4. Discussion

In this study, we investigated whether HRCT abnormalities could predict the occurrence of AE-IPF in the independent two ethnic cohorts by using the same definition criteria of AE-IPF, and demonstrated that a larger extent of GGO and fibrosis area on HRCT and Japanese ethnicity were independent risk factors for AE-IPF. 

Areas of minor GGO are findings frequently seen on HRCT in patients with IPF [11,12]. They reflect various pathological abnormalities, such as inflammation, fibrotic changes, honeycomb cysts with secretions, and superimposed diffuse alveolar damage [22]. The recent statements on IPF emphasized that the discrimination of “pure” GGO from GGO associated with traction bronchiectasis or other features of fibrosis (i.e., non-pure GGO) was very important for identifying IPF and AE-IPF [23]. Although we did not distinguish between these two types of GGO in this study, we excluded patients with AE-IPF at the time of HRCT with the most careful attention. Additionally, a recent study demonstrated that GGO strongly correlated with the fibrotic regions on micro-CT [24]. Although no studies previously investigated whether the area of GGO was predictive of AE-IPF, the areas of GGO in this study could have been associated with the potential of fibrosis and the risk of AE-IPF. It could be an interesting topic to evaluate the differences in the clinical features, risk of AE-IPF, and survival between patients with pure GGO and those with non-pure GGO. In this study, however, we carefully excluded patients with AE-IPF at enrolment, and the majority of patients presented with non-pure GGO. What we found was that the area of non-pure GGO at enrolment was associated with the subsequent occurrence of AE-IPF.

Previous studies showed that the extent of fibrosis on HRCT was correlated with the extent of fibrotic areas on histopathology. Fibrotic areas, including reticular opacity and honeycombing, were found to be predictors of mortality in patients with IPF [11,12,13]. Considering that AE-IPF is one of the most common causes of death in patients with IPF, it is likely that the extent of fibrosis is associated with the risk of AE-IPF. Data from literature are scarce and conflicting. Whereas Suzuki et al. reported that the extent of fibrosis on HRCT was shown to predict postoperative AE-IPF in patients with lung cancer and IPF [25], Mura et al. did not find that the extent of fibrosis on HRCT was a significant predictor of AE-IPF [4]. They evaluated the HRCT findings by using a rough five-point scale which might have affected their results. In our study, we applied a more precise quantification of HRCT extent by using a 5% scale which may be the reason why we came to different results and could demonstrate that a larger extent of GGO and fibrosis areas on HRCT are risk factors for AE-IPF. 

AE may occur at any time during the course of IPF, although it appears to be more common in patients with more advanced IPF. Previous studies demonstrated that low forced vital capacity (FVC) [3], low DLco [4], high alveolar to arterial oxygen pressure difference [7], and a high Medical Research Council dyspnea scale [6] are risk factors of AE. Collard et al. showed that a low baseline FVC and baseline supplemental oxygen were strongly associated with the risk of AE (hazard ratio 0.67 and 2.47, respectively) [5]. In addition, pulmonary hypertension was also associated with the subsequent development of AE with the hazard ratio of 2.51 [9]. Taken together, advanced IPF appears to be a significant risk factor of AE, and this is supported by our study, which demonstrated that larger extents of interstitial abnormalities on HRCT are significantly associated with AE. However, our data cannot completely clarify whether AE-IPF occurs more frequently in patients with more severe disease, or whether AE-IPF are just diagnosed more frequently in more severe patients because they are more likely to be hospitalized when their worsening symptoms are more pronounced due to greater impairment in the advanced disease stages. 

The presence of emphysema may complicate the evaluation of HRCT findings and also affect the outcome of patients with IPF. Patients with combined pulmonary fibrosis and emphysema (CPFE) have more frequent secondary pulmonary hypertension associated with high mortality [26]. However, the impact of emphysema on the incidence of AE-IPF varies, according to different studies. Some studies suggested that concomitant emphysema is a significant risk factor of AE-IPF [8,12], while Sugino et al. demonstrated that, during a three-year follow-up, the incidence of AE-IPF was significantly lower in patients with CPFE than in patients with IPF alone (10.5% vs. 28.1%; *p* = 0.012) [27]. In our study, the extent of emphysema was not associated with the risk of AE-IPF. These discrepancies may be related to different definitions of concomitant emphysema and to the wide variation of the incidence of concomitant emphysema ranging from 8% to 42% in these studies. Further investigations are needed to clarify the impact of concomitant emphysema on AE-IPF.

We demonstrated that the Japanese ethnicity was independently associated with the risk of AE-IPF. This is in contrast with data from the INPULSIS trials, which showed that there was no difference in the risk of AE-IPF between Asian and White patients [15] but was in agreement with Reichmann et al., who demonstrated that Caucasian patients with IPF experienced less AE-IPF than non-Caucasian patients with IPF [8]. Previous studies showed that the impact of genetic risk factors of IPF was different in different ethnicities [14,28,29]. The MUC5B polymorphism was significantly associated with disease susceptibility and prognosis in Caucasian and Mexican patients with IPF [29,30,31], but this association was relatively weak in Asian patients with IPF [29,32]. Further studies are needed to definitely clarify the genetic factors contributing to the risk of AE-IPF. 

Our study has several potential limitations. Firstly, the design of this study was retrospective, and the treatments were not identical in the two populations with regard to the use of prednisone and azathioprine [8]. Although the univariate analysis demonstrated no significant associations between the use of corticosteroids or immunosuppressants and the occurrence of AE-IPF, we cannot completely exclude the potential effects of these treatment regimens on the occurrence of AE-IPF. Regarding pirfenidone treatment, the effect of this drug on the incidence of AE-IPF was not significant in the univariate analysis. Recent trials showed that nintedanib can reduce the risk of AE-IPF [33,34]. Since our cohorts were collected between 2003 and 2012, nintedanib was not administered to the patients studied. However, we cannot exclude potential effects of the different treatment regimen on the differences in CT appearances, such as GGO and fibrosis. Secondly, the number of patients included in this study was not large. Because the incidence of AE-IPF was low, further prospective investigations with a larger sample size and higher event rate are necessary to confirm the current results. Thirdly, we cannot completely exclude the effects of potential confounding environmental factors. Johannson et al. showed that exposure to increased concentrations of the air pollutants, ozone and nitrogen dioxide, was associated with AE-IPF [35]. Fourthly, automated quantification of HRCT features was not applied in this study. Rosas et al. demonstrated the utility of an automated quantification program for evaluating the areas of characteristic features for interstitial lung diseases [36]. This procedure could be more objective compared with our visual assessment, and should be explored in future studies. Fifthly, we could not perform bronchoalveolar lavage (BAL) in all patients. We agree with the essential role of BAL for excluding pulmonary infection; however, because of the risk of promoting respiratory insufficiency in patients with suspected AE-IPF, we did not perform BAL in all patients with suspected AE-IPF. We carefully excluded pulmonary infection by using the results of sputum culture, serum antigens, urine antigens, and pharyngeal swab antigens. Collard et al. demonstrated that the prognosis of suspected AE-IPF was similar as of definite AE-IPF [37]. Therefore, we believe it was reasonable to include patients with suspected AE-IPF without BAL examination in this study.

Despite these limitations, the results from our study appear to be important in that we can demonstrate that the extent of lung parenchymal abnormalities on HRCT and Japanese ethnicity are independent risk factors of AE-IPF.

In conclusion, the extent of lung parenchymal abnormalities on HRCT and Japanese ethnicity appear to be independently associated with the incidence of AE-IPF. HRCT may be useful for assessing the risk of AE-IPF in patients with IPF, independent of ethnicity.

## Figures and Tables

**Figure 1 jcm-08-02069-f001:**
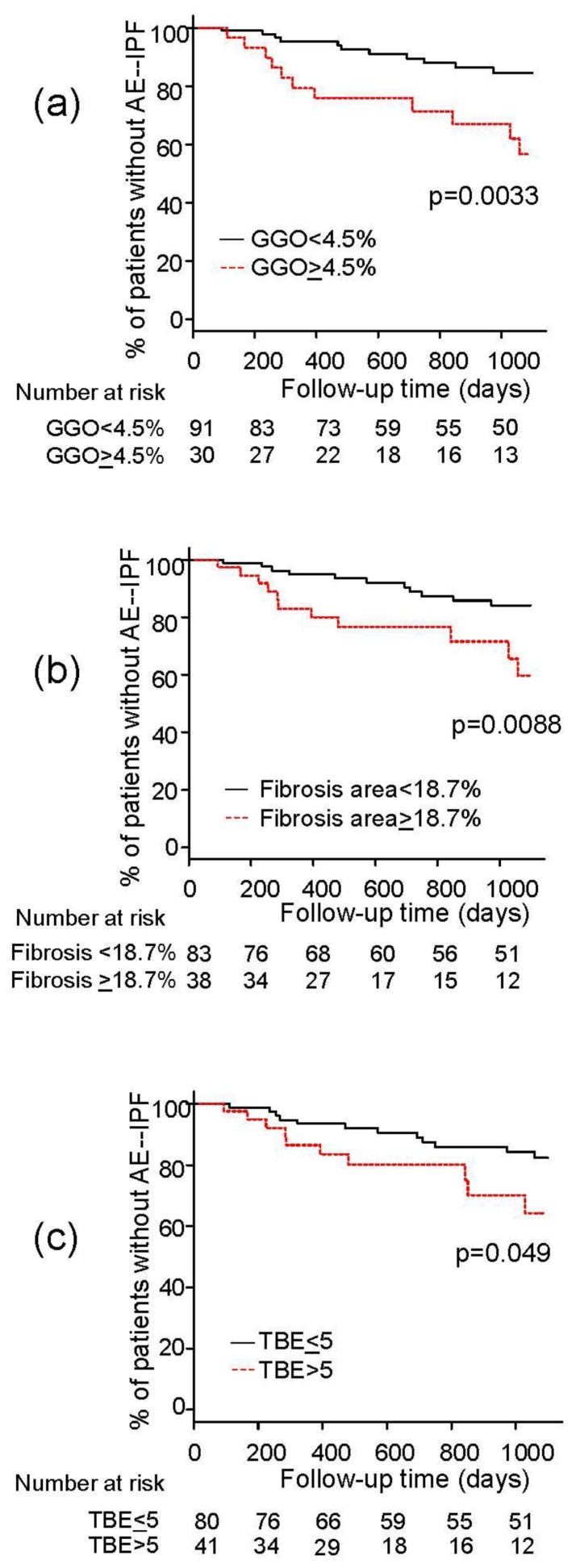
Kaplan–Meier analysis for the onset of acute exacerbation of idiopathic pulmonary fibrosis (AE-IPF) grouped by the extent of (**a**) ground glass opacity (GGO), (**b**) fibrosis area, and (**c**) traction bronchiectasis (TBE).

**Table 1 jcm-08-02069-t001:** Clinical characteristics of the study subjects.

	All	German	Japanese	*p* Value	Missing Values
Number of the subjects	121	54	67		
Age (years)	68.5 ±7.6	68.1 ± 7.6	68.9 ± 7.5	NS	0
Gender (male/female)	98/23	41/13	57/10	NS	0
Smoking (Cu or Ex/Non)	86/30	33/16	53/14	NS	5(4.1%)
VC (percent predicted)	75.0 ± 16.5	74.7 ±14.9	75.3 ± 18.1	NS	0
DL_CO_ (percent predicted)	45.5 ± 15.5	45.8 ±17.3	45.4 ± 14.2	NS	6(5.0%)
Use of corticosteroid (yes/no)	57/64	41/13	16/51	<0.01	0
Use of immunosuppressive agent (yes/no)	30/91	27/27	3/64	<0.01	0

Data are shown as mean ± SEM. IPF, idiopathic pulmonary fibrosis: Cu, current smoker; Ex, ex-smoker; Non, non-smoker; VC, vital capacity; DL_CO_, diffusing capacity of the lung for carbon monoxide; NS, not significant.

**Table 2 jcm-08-02069-t002:** HRCT findings in German and Japanese patients with idiopathic pulmonary fibrosis.

HRCT-Findings	German Patients	Japanese Patients	*p* Value
GGO (%)	2.92 ± 4.11	3.85 ± 5.12	0.015
Fibrosis (%)	16.60 ± 7.99	12.80 ± 6.68	<0.01
Emphysema (%)	1.72 ± 3.63	4.02 ± 8.62	0.010
Zones with traction bronchiectasis (n)	5.19 ± 1.13	4.27 ± 1.61	<0.01

Statistical significance was tested by the Mann–Whitney *U*-test. HRCT, high-resolution computed tomography: IPF, idiopathic pulmonary fibrosis: GGO, ground glass opacity.

**Table 3 jcm-08-02069-t003:** Univariate Cox-proportional hazards analysis for predicting acute exacerbation of idiopathic pulmonary fibrosis (AE-IPF).

Variables	HR	95%CI	*p* Value
Age (continuous)	0.99	0.94–1.05	0.73
Male sex	0.87	0.29–2.57	0.80
Positive smoking history	0.97	0.36–2.64	0.96
Pack-year	0.99	0.97–1.01	0.19
Japanese Ethnicity	3.17	1.07–9.37	0.037
%VC ≥ 74% (median)	0.37	0.15–0.92	0.031
%DLco ≥ 44% (median)	0.51	0.21–1.23	0.13
Use of corticosteroid	1.76	0.76–4.08	0.19
Use of immunosuppressive agent	1.03	0.35–3.06	0.96
Use of pirfenidone	0.52	0.16–1.77	0.30
GGO > 4.5%	3.27	1.42–7.55	0.0055
Fibrosis > 18.7%	2.93	1.26–6.79	0.012
Emphysema > 8.3%	1.67	0.56–4.92	0.36
Zones with traction bronchiectasis >5 (n)	2.29	0.98–5.34	0.055

Statistical significance was tested by the Cox-proportional hazards model. AE-IPF, acute exacerbation of idiopathic pulmonary fibrosis: HR, hazard ratio: VC, forced vital capacity: DL_CO_, diffusing capacity of carbon monoxide: GGO, ground glass opacity: CI, confidence interval.

**Table 4 jcm-08-02069-t004:** Multivariate Cox-proportional hazards analysis for predicting AE-IPF.

Variables	HR	95%CI	*p* Value
Age (continuous)	0.97	0.91–1.03	0.29
Male sex	0.59	0.18–1.89	0.37
Japanese Ethnicity	4.59	1.50–13.98	0.0074
%VC > 74% (median)	0.55	0.21–1.43	0.22
GGO > 4.5%	2.81	1.13–6.96	0.026
Fibrosis area > 18.7%	2.68	1.06–6.76	0.037

AE-IPF, acute exacerbation of idiopathic pulmonary fibrosis: HR, hazard ratio: VC, forced vital capacity: GGO, ground glass opacity: CI, confidence interval.

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
