# Peer review of "Baseline High-Resolution CT Findings Predict Acute Exacerbation of Idiopathic Pulmonary Fibrosis: German and Japanese Cohort Study"

_jcm, 2019, doi:10.3390/jcm8122069_

Round 1

Reviewer 1 Report

Major Comments:

The study concludes that there were significant clinical differences between the German and Japanese cohorts. However, there were significant differences between the treatments in these groups: a greater proportion of the German patients received corticosteroid or immunosuppressive therapy (and, when appropriately randomised as in INPULSIS, the risk of AE-IPF was comparable). Given the differences in CT appearance (with more GGO in the Japanese patients and more fibrosis and traction bronchial dilation in the German, can the authors be confident that both cohorts could be diagnosed with IPF? Could the treatment differences account for the differences in CT appearance and with the incidence of AE-IPF? Further discussion of these issues should be added to the Discussion section on page 14, when the study’s limitations are mentioned.

Could the findings that associate greater GGO, fibrosis and lower VC with AE-IPF merely be representative that AE-IPF is either more frequent as the disease becomes more severe, or is more frequently diagnosed as the disease becomes more severe, since the patients are more likely to present when their worsening symptoms are more evident due to greater impairment in the later stages of disease (and this is implied by the studies referenced in page 13 ie 3,4,5,6,7 and 9)? This should be explored further in the discussion with reference to the diagnostic criteria used to define AE-IPF.

Do the authors consider that, since GGO was the best CT predictor of AE-IPF, they should reclassify their cases using the two types of GGO referred to in page 13 of their Discussion section? Please explain why this was not performed, since it will be important in the interpretation of the study data.

Minor comments:

‘Traction bronchial dilatation’ is a preferable radiological term to ‘traction-bronchiectasis’.

Page 2 line 60: It may be inaccurate to state that the clinical course of IPF is invariably progressive – there are cases that seem to stabilise over many years (assuming that the diagnosis of IPF in these cases is correct). It may be better to rephrase this sentence (ie 'never say never').

Why was an automated software programme not utilised to quantify features of interstitial lung disease?

Page 3 line 115: the sentence should be altered to – ‘Due to the generation of CT scanners used / utilised..’

Author Response

Reviewer1

Major Comments:

Comment 1 (C1):   The study concludes that there were significant clinical differences between the German and Japanese cohorts. However, there were significant differences between the treatments in these groups: a greater proportion of the German patients received corticosteroid or immunosuppressive therapy (and, when appropriately randomized as in INPULSIS, the risk of AE-IPF was comparable). Given the differences in CT appearance (with more GGO in the Japanese patients and more fibrosis and traction bronchial dilation in the German, can the authors be confident that both cohorts could be diagnosed with IPF? Could the treatment differences account for the differences in CT appearance and with the incidence of AE-IPF? Further discussion of these issues should be added to the Discussion section on page 14, when the study’s limitations are mentioned

Response 1 (R1):     Thank you very much for your helpful comments. We are confident that both cohorts were diagnosed with IPF. The presence of minor GGO is not an inconsistent finding for the diagnosis of IPF. As a matter of fact, the mean proportion of GGO in this study was minor indeed, with only around 3% in both cohorts (2.92% vs 3.85%), and the major HRCT findings were honeycombing and fibrosis, which is consistent with the diagnosis of IPF, independently confirmed by the two radiologists. Regarding your concern that treatment differences may have accounted for the differences in CT appearance and with the incidence of AE-IPF, the patient cohort in this study was collected before 2012, and no patients have been treated with nintedanib, which can enable us to exclude the potential effect of this medication on the occurrence of AE-IPF. The use of corticosteroids or immunosuppressant was not associated with the occurrence of AE-IPF in the univariate analysis. Based on a comment from Reviewer 2 we also looked at pirfenidone treatment and found no significant effect of this drug on the incidence of AE-IPF in the univariate analysis. However, since we cannot completely exclude the potential effects of the treatment regimen on the occurrence of AE-IPF, we have added this drawback in the limitation section.
We agree that we cannot exclude the potential effects of the different treatment regimen on the differences in CT appearances  We added this drawback in the limitation section.

C2:      Could the findings that associate greater GGO, fibrosis and lower VC with AE-IPF merely be representative that AE-IPF is either more frequent as the disease becomes more severe, or is more frequently diagnosed as the disease becomes more severe, since the patients are more likely to present when their worsening symptoms are more evident due to greater impairment in the later stages of disease (and this is implied by the studies referenced in page 13 ie 3,4,5,6,7 and 9)? This should be explored further in the discussion with reference to the diagnostic criteria used to define AE-IPF.

R2:      We completely agree with your comment. Our data cannot address whether AE-IPF are occurring more frequently in patients with more severe disease, or AE-IPF are just diagnosed more frequently in severe disease because patients with more severe disease are more likely to visit the hospital. We added this consideration in the 4th paragraph of the Discussion section.

C3:      Do the authors consider that, since GGO was the best CT predictor of AE-IPF, they should reclassify their cases using the two types of GGO referred to in page 13 of their Discussion section? Please explain why this was not performed, since it will be important in the interpretation of the study data.

R3:      We appreciate your constructive comment. The difference in pure GGO and non-pure GGO (i.e. GGO associated with traction bronchiectasis or fibrosis) could be a novel indicator for differentiating AE-IPF from other features of IPF. Therefore, it can be an interesting topic to evaluate the difference in the clinical features, risk of AE-IPF, and survival between patients with pure GGO and those with non-pure GGO. However, we have carefully excluded patients with AE-IPF at enrollment, hence the majority of patients presented non-pure GGO in this study. What we have found in this study was that the area of non-pure GGO at the enrollment was associated with the subsequent occurrence of AE-IPF. We have added this to the 2nd paragraph of the Discussion section.

Minor comments:

C4:      Traction bronchial dilatation’ is a preferable radiological term to ‘traction-bronchiectasis’.

R4:      We prefer to keep the wording traction bronchiectasis because this term has consistently been used in the context of IPF and ILD in previous papers, especially also in the recent Diagnosis of IPF Statement/Guidelines publications defining the new category of the probable UIP pattern. The ILD community is very familiar with this term.

C5:      Page 2 line 60: It may be inaccurate to state that the clinical course of IPF is invariably progressive – there are cases that seem to stabilize over many years (assuming that the diagnosis of IPF in these cases is correct). It may be better to rephrase this sentence (ie 'never say never').

R5:      Thank you very much for your suggestion. We rephrased this sentence. (page 2, line 60).

C6:      Why was an automated software program not utilized to quantify features of interstitial lung disease?

R6:      Thank you very much for your constructive suggestion. This is a very current topic in radiological evaluation of interstitial lung disease (ILD). Although promising, softwares to quantify fibrosis and GGO in the clinical practice are not yet widely used and validation studies are ongoing. Therefore , we chose analogue evaluation by two expert radiologists, in our opinion reflecting the most widely used practice. However, since your suggestion is highly appropriate, we have added this as a further limitation in the Discussion section.

C7:      Page 3 line 115: the sentence should be altered to – ‘Due to the generation of CT scanners used / utilised..’

R7:      Thank you very much for your comment. We have modified the sentence accordingly.

Reviewer 2 Report

The paper proposed by Hirano et al. entitled "Baseline high-resolution CT findings predict acute 2 exacerbation of idiopathic pulmonary fibrosis: 3 German and Japanese cohort study" is well-written and very interesting, as it focused on the prediction of AE in IPF through radiological evaluation, that is one of the most debated topics of this disease. 

Results ar interesting and in line with other previous works published on the field. I have just some minor concerns:

Methods: radiological assessment is not completely described. Quantification of radiological features was made through visual assessment? Please clarify.

Results: BAL was performed in all patients with AE to exclude lung infection?

Results: was any patient taking pirfenidone during the study? Please clarify this aspect and, in case of a relevant number of patients taking prifenidone, enlarge statistical analysis and  discussion on this topic 

Author Response

Reviewer2

Comment 8 (C8):    Methods: radiological assessment is not completely described. Quantification of radiological features was made through visual assessment? Please clarify.

Response 8 (R8):      Thank you very much for your comment. As the reviewer 1 has also suggested, one of the limitations of this study was the analogue evaluation of HRCT by the two radiologists. Automated quantification program could be an alternative procedure for enabling us a more objective evaluation of HRCT features. We have modified the Methods and Limitation sections.

C9:      Results: BAL was performed in all patients with AE to exclude lung infection?

R9:      Thank you very much for your practical comment. We agree with the essential role of BAL for excluding pulmonary infection. However, since we were worried to induce respiratory insufficiency in patients with suspected AE-IPF, we did not perform BAL in all patients with suspected AE-IPF. We have carefully excluded pulmonary infection by using the results of sputum culture, serum antigens, urine antigens, and pharyngeal swab antigens. Collard et al. (2013) have demonstrated that the prognosis of suspected AE-IPF was as similar as definite AE-IPF. Therefore, we believe it can be reasonable to include patients with suspected AE-IPF without BAL examination as those with definite AE-IPF. However, we have added this drawback as a limitation in the manuscript.

C10:    Results: was any patient taking pirfenidone during the study? Please clarify this aspect and, in case of a relevant number of patients taking pirfenidone, enlarge statistical analysis and discussion on this topic

R10:    Thank you very much for your valuable comment. Our patient cohort included a total of 25 patients (German, 6; Japanese, 19) that have taken pirfenidone during the study period. In the univariate analysis, use of pirfenidone was associated with a hazard ratio of 0.52 (95%CI, 0.16-1.77, p=0.30) for the occurrence of AE-IPF. Unfortunately the dose, timing to start, treatment duration for pirfenidone, and the timeframe between pirfenidone start and the occurrence of AE-IPF widely varied among patients, so that we cannot draw any conclusion on the impact of pirfenidone on the occurrence of AE-IPF in our cohort. We have added the data in the Result section and Table 3, and this drawback in the limitation section.